# Heritability of Gene Expression Measured from Peripheral Blood in Older Adults

**DOI:** 10.3390/genes15040495

**Published:** 2024-04-16

**Authors:** Sri C. Kanchibhotla, Karen A. Mather, Nicola J. Armstrong, Liliana G. Ciobanu, Bernhard T. Baune, Vibeke S. Catts, Peter R. Schofield, Julian N. Trollor, David Ames, Perminder S. Sachdev, Anbupalam Thalamuthu

**Affiliations:** 1Centre for Healthy Brain Ageing, Discipline of Psychiatry & Mental Health, School of Clinical Medicine, Faculty of Medicine and Health, University of New South Wales, Sydney, NSW 2052, Australia; 2Neuroscience Research Australia, Sydney, NSW 2031, Australia; 3Department of Mathematics and Statistics, Curtin University, Perth, WA 6845, Australia; 4Discipline of Psychiatry, Adelaide Medical School, The University of Adelaide, Adelaide, SA 5005, Australia; 5Department of Psychiatry, University of Münster, 48149 Münster, Germany; 6Department of Psychiatry, Melbourne Medical School, The University of Melbourne, Melbourne, VIC 3052, Australia; 7The Florey Institute of Neuroscience and Mental Health, The University of Melbourne, Melbourne, VIC 3052, Australia; 8School of Medical Sciences, University of New South Wales, Sydney, NSW 2052, Australia; 9Department of Developmental Disability Neuropsychiatry, Discipline of Psychiatry & Mental Health, School of Clinical Medicine, Faculty of Medicine and Health, University of New South Wales, Sydney, NSW 2052, Australia; 10Academic Unit for Psychiatry of Old Age, University of Melbourne, St George’s Hospital, Kew, Melbourne, VIC 3010, Australia; 11National Ageing Research Institute, Parkville, VIC 3052, Australia; 12Neuropsychiatric Institute, Euroa Centre, Prince of Wales Hospital, Sydney, NSW 2031, Australia

**Keywords:** gene expression, older adults, twins, heritability

## Abstract

The contributions of genetic variation and the environment to gene expression may change across the lifespan. However, few studies have investigated the heritability of blood gene expression in older adults. The current study therefore aimed to investigate this question in a community sample of older adults. A total of 246 adults (71 MZ and 52 DZ twins, 69.91% females; mean age—75.79 ± 5.44) were studied. Peripheral blood gene expression was assessed using Illumina microarrays. A heritability analysis was performed using structural equation modelling. There were 5269 probes (19.9%) from 4603 unique genes (23.9%) (total 26,537 probes from 19,256 genes) that were significantly heritable (mean h^2^ = 0.40). A pathway analysis of the top 10% of significant genes showed enrichment for the immune response and ageing-associated genes. In a comparison with two other gene expression twin heritability studies using adults from across the lifespan, there were 38 out of 9479 overlapping genes that were significantly heritable. In conclusion, our study found ~24% of the available genes for analysis were heritable in older adults, with only a small number common across studies that used samples from across adulthood, indicating the importance of examining gene expression in older age groups.

## 1. Introduction

Despite the worldwide ageing of the population, our understanding of the underlying molecular processes that contribute to human ageing is sparse. Chronological ageing is a significant risk factor for many complex diseases and conditions, including cancer and late-onset Alzheimer’s disease. To develop effective strategies to prevent and treat age-related diseases, it is imperative to increase our understanding of the biology of human ageing.

Gene expression is influenced by genetic, environmental, and stochastic factors and it may vary by tissue type [1,2,3]. Heritability studies have been used to elucidate the importance of genetic and environmental factors on gene expression. The importance of these factors may change across the lifespan. Yet few studies have explicitly studied heritability of gene expression at older ages.

The most commonly used tissue for gene expression heritability studies is peripheral blood, as it is easily accessible and relatively inexpensive to collect. The mean heritability (h^2^) of blood gene expression has been reported as low to moderate (h^2^—0.10 to 0.32). The previous studies have examined adolescents or samples from across adulthood and varied in sample sizes from 100 to ~2700 [4,5,6,7]. Most studies have used microarrays to assess gene expression. In a family study, Price et al. (2011) observed low mean h^2^ (0.15) for gene expression in individuals aged 18 to 85 years (*n*~1000). Using a sample of adolescent twins (*n*~100), Powell and colleagues (2012) found a mean heritability estimate of 0.32. In a large study of 2752 twins aged 18–65 years, the authors reported 777 genes with significant but low mean h^2^ (0.10) [7]. In the Consortium for the Architecture of Gene Expression (CAGE) twin and family study, low to moderate mean h^2^ (19%) of gene expression was observed [8]. Further, in another large study [459 monozygotic (MZ) and 150 dizygotic (DZ) twin pairs, mean age ± SD = 36.7 ± 14.0] that had an overlapping sample with the aforementioned microarray study by Wright et al., gene expression was assessed using RNA sequencing, with a mean h^2^ of 20% observed. SNP heritability of gene expression was also estimated at 26%, with both measures of heritability (SNP and twin-based) being highly correlated [4].

To date, only one very small study (total *n* = 12) has reported the heritability of gene expression specifically in older adults (70+ years—females only) observing higher intra-class correlations in MZ twin pairs compared to DZ for genes predominantly involved in antigenic immune and defence responses [9]. Therefore, the current twin study aims to evaluate and describe the heritability of blood gene expression in a larger sample of older adults aged 69 to 94 years.

## 2. Methods

### 2.1. Participants

The Older Australian Twin Study (OATS) is a longitudinal study of twins and their siblings aged 65 years and above that were recruited via Twins Research Australia and through recruitment drives across three Eastern states of Australia—New South Wales, Queensland, and Victoria. Inclusion criteria were having a consenting co-twin and the ability to complete the questionnaires in English, whereas individuals diagnosed with any progressive malignancy or other life-threatening illness/acute psychotic disorder were excluded from the study. The full details of the study have been published elsewhere [10]. The zygosity of the sample was assessed based on identity by descent using available genome-wide genotyping data along with self-report data [11]. Demographic, medical, and health information were obtained through an extensive interview and fasting peripheral blood samples were donated for genetic and biochemical tests, including white blood cell count.

The study was approved by human research ethics committees for the Australian Twin Registry, University of New South Wales, University of Melbourne, Queensland Institute of Medical Research, and the South-eastern Sydney and Illawarra Area Health Service and written informed consent was provided by all participants.

The current study comprises 246 OATS participants (52 DZ pairs, 71 MZ pairs) with available gene expression data. The mean age of the sample used in this study was 75.79 years (SD = 5.44) with ages ranging from 69.4 to 93.5 years. All participants were of European descent.

### 2.2. Gene Expression Analysis

Overnight fasting blood samples for RNA extraction were collected in PAX gene tubes (PreAnalytiX, QIAGEN, Hombrechtikon, Switzerland). Total RNA from blood was extracted using the PAX gene Blood miRNA kit (PreAnalytiX, QIAGEN). Successfully extracted RNA samples with a RNA integrity number (RIN) ≥ 6 were used for subsequent gene expression assays [12]. The Illumina Whole-Genome Gene Expression Direct Hybridization Assay System HumanHT-12 v4 array (Illumina Inc., San Diego, CA, USA) was used to assay gene expression following the standard manufacturer’s protocol [13]. The intensity values of the raw gene expression were extracted from Genome Studio (Illumina). The Bioconductor R package limma [14] was used for the pre-processing of the raw expression data including quality control (QC) and normalization. The pre-processing steps included checking of the sample outliers through multidimensional scaling plots, background correction and quantile normalization using the neqc function in the limma package. Of the total probes on the array (*n* = 47,323), only those expressed (detection *p*-value < 0.05) in at least 3 samples and well annotated (*n* = 26,537) were retained for heritability analysis. The final list of probes had no missing values.

### 2.3. Statistical Analysis and Bioinformatics

#### 2.3.1. Heritability

The influence of genetic and environmental factors on gene expression levels was studied using the classical twin method [15]. MZ twins share all genes, while Z twins share only 50% of their genes. The effect of genetic (A), shared (C), and non-shared (E) environmental factors on gene expression was estimated by comparing the similarity between the identical (MZ) and non-identical (DZ) twin pairs. If the twin correlation between the MZ pairs is more than the DZ pairs, it indicates the influence of genetic factors (A). Similarly, if the DZ twin correlation is more than half the MZ correlation, it indicates the influence of shared environmental (C) factors.

Heritability analysis was performed using structural equation modelling in Open Mx [16,17] based on the standard ACE model. For each probe, the maximum likelihood ratio, Akaikes information criteria (AIC) (Akaike, 1987), and the *p*-value were assessed to determine the best fitting model by comparing the ACE model to the AE, CE, and E models. The *p*-values for heritability under the AE model were obtained by comparing the likelihoods of the AE and E models.

The covariates included in the analyses were age, sex, assay batch, RNA integrity number (RIN), and total white blood cell count (includes eosinophils, lymphocytes, basophils, neutrophils, and monocytes). Heritability was calculated using the rank-based inverse normal transformed (Rank Norm function in the R package RNOmni) [18] residuals under a regression model after adjusting for the covariates listed above. Two estimates of gene expression heritability were calculated: (i) at the individual probe level and (ii) at the gene level, if multiple probes were available within a gene. The gene heritability was defined as the maximum value of probe heritabilities. Adjusted *p*-values (FDR) were calculated using Benjamini–Hochberg procedure as implemented in the R package function p. adjust (v4.0; R Core Team 2020; https://www.R-project.org/). The Bootstrapping confidence intervals were also calculated using bootstrapCI function in Open Mx. The Manhatton plot of the gene heritability *p*-values was done using the R package CMplot (LiLin-Yin (2024). CMplot: Circle Manhattan Plot).

#### 2.3.2. Gene Set Analysis

Gene set analysis was performed using GENE2FUNC procedure in FUMA (Functional Mapping and Annotation of Genome-Wide Association Studies) [19]. This tool tested the representation of a given set of genes in different functional gene sets, including GO terms, using the background set of all genes from the OATS heritability results (*n* = 19,256). FUMA was also used to assess the expression pattern of a list of genes in various tissues. FDR (Benjamini–Hochberg false discovery rate) was used to control for multiple hypotheses testing.

#### 2.3.3. Correlations between Gene Expression Heritability with Gene Length and Percentage GC Content

Pearson’s correlations between the heritability of gene expression for the FDR-significant genes (FDR < 0.05) with the percentage GC content and gene length were estimated. The gene length and percentage GC content were downloaded from the Bio mart-Ensembl browser.

#### 2.3.4. Ageing and Longevity Enrichment Analysis

Enrichment analysis of ageing-associated and longevity-related genes was conducted for significant genes (FDR < 0.05). The human ageing-associated and longevity-related genes were downloaded from the Gen Age database [https://genomics.senescence.info, downloaded in August 2021] [20]. Enrichment analysis was performed using Fisher’s test in R Statistical Software version 4.0.0.

#### 2.3.5. Overlap Analysis for the Heritability of Gene Expression across Three Studies

This analysis used the h^2^ results from the current study (OATS) and data from two other published studies from the Netherlands [4,7]. The two Dutch studies are only partially independent, as they share 60% of their samples but used different gene expression methodologies, microarray [7], and RNA sequencing [4]. The total numbers of genes available from the three studies were *n* = 19,256 (OATS), *n* = 18,392 (Wright et al. [7]), and *n* = 11,409 (Ouwens et al. [4]). The overlap of genes between the three cohorts based on Ensemble IDs was 9479 [21]. To ensure consistency when comparing the results across the three studies, new FDR values were generated for the overlapping set of genes across the three cohorts and the common significantly heritable genes across the cohorts were identified.

The list of common heritable genes across the three studies were then checked for prior associations with traits in the GWAS catalogue (NHGRI-EBI Catalog of human genome-wide association studies). In addition, an enrichment analysis was also conducted in FUMA, using the background set of all genes from the OATS heritability results (*n* = 19,256), to identify the traits and pathways enriched in this gene set. Furthermore, we checked if any of these common heritable genes overlap with the set of ageing- associated and longevity- genes.

## 3. Results

### 3.1. Demographics

The demographic details of the sample are shown in Table 1. There were more females (69.91%) than males. The mean age of the sample was 75.79 (SD ± 5.44). The mean BMI was 27.63 (SD ± 4.74) and the mean WBC count of the sample was 6.41 (SD ± 1.66). There were a total of eight (3.25%) current smokers in the sample. A total of 77 individuals (31.30%) reported drinking more than 30 standard drinks per month. There were no significant differences between MZ and DZ pairs for age (*p* = 0.537), sex (*p* = 0.402), BMI (*p* = 0.764), and WBC count (0.392).

### 3.2. Heritability Analysis

Heritability (h^2^) analysis was performed using ACE and other sub-models. The AE model was the best fitting model (based on AIC and *p*-values of ACE vs. AE) for the majority of probes (98.2%), which is consistent with previous heritability analyses performed in the OATS cohort, where we found that AE was the model of best fit when examining neuroimaging and cognitive phenotypes [22,23]. Hence, both gene and probe heritability were summarized using the AE model.

#### 3.2.1. Probe Heritability

Heritability analyses examined a total of 26,537 probes from 19,256 genes. Of these probes, 5269 probes (19.86%) from 4603 unique genes were significant (FDR < 0.05) (Appendix A). The bootstrap confidence intervals for the FDR-significant probes were also calculated giving consistent results with the confidence intervals calculated using statistical approximations. The mean heritability across all the significant probes (*n* = 5269) was moderate (h^2^ = 0.40; SD = ±0.10; range = 0.25–0.87). Of the FDR-significant probes, the highest heritability (h^2^ = 0.87) was observed for an *ERAP2* transcript (ILMN_1743145, FDR =2.91 × 10^−22^), while the lowest heritability (h^2^ = 0.27) was observed for a *BRD9* transcript (ILMN_1651405, FDR =0.0499). Table 2 lists the top 10 heritable probes.

At a nominally significant level (*p*-value < 0.05), there were 7750 (29.20%) heritable probes from 6603 unique genes. The mean heritability of the nominally significant probes (*p*-value < 0.05) (*n* = 7750) was also low to moderate (h^2^ = 0.35 ± 0.10).

#### 3.2.2. Gene Heritability

The Manhattan plot for the twin-based heritability for 19,256 genes is shown in Figure 1. Of these genes, 4603 (23.90%) were significant (FDR < 0.05). Similarly to the probe heritability results, the mean heritability of FDR-significant genes was found to be moderate (h^2^ = 0.40).

There were 6598 unique genes (34.26% of 19,256 genes available) that were heritable at a nominally significant level (*p*-value < 0.05) with low to moderate mean heritability (h^2^ = 0.35).

Of the 4603 FDR-significant genes, 576 had multiple probes ranging from 2 to 5 per gene. Four genes (*LAMP2*, *METRNL, TYMP*, and *SLX1B-SULT1A4*) had a maximum of five of significant probes each. A chromosome-wise analysis of the FDR-significant genes is shown in Appendix A, with chromosome 22 having the highest proportion of FDR-significant genes (FDR-significant genes/total number of genes available for a specific chromosome). However, the average heritability was highest for genes located on chromosome 20 followed by chromosome 22 (Appendix A).

### 3.3. Correlations between Gene Expression Heritability with Gene Length and Percentage GC Content

Pearson’s correlation was calculated between the heritability of the gene expression of the FDR-significant genes (*n* = 4430 with data available) with percentage GC content and gene length (Appendix A). A significant positive correlation was found for percentage GC content (r = 0.093, *p* = 5.175 × 10^−10^), whereas gene length was not significant (r = −0.022, *p* = 0.146).

### 3.4. Gene Set Analysis

Gene set analysis was performed on the top 10% of FDR-significant genes (*n* = 460) using the Gene2Func procedure in FUMA. There were 19 significant GO biological processes and five significant canonical pathways. The top 10 significant GO biological processes and canonical pathways are shown in Table 3. As expected, given the tissue source, genes related to immune response were in the top GO biological processes and canonical pathways. Tissue-specific analysis in 30 general tissues indicated that this gene set is significantly enriched in three tissues including blood, spleen, and lungs (Appendix A).

### 3.5. Ageing and Longevity Enrichment Analysis

The human ageing- and longevity-related gene lists were downloaded from GenAge (see GenAge: The Ageing Gene Database (https://genomics.senescence.info/genes/index.html, downloaded in August 2021)), has 357 genes, of which 132 were missing from the OATS gene list (*n* = 19,256), making a total of 225 longevity-related genes available for enrichment analysis. Similarly, there were 307 ageing-associated genes, of which 32 were omitted from analysis, making a total of 265 ageing associated genes available for enrichment analysis (Appendix A). Fifty-five genes overlap between these two lists. The OATS FDR-significant gene set showed enrichment for both longevity (Fisher’s exact test *p*-value = 7.702 × 10^−5^; OR= 1.77; 95% CI (1.33, 2.35)) and the ageing-associated genes (Fisher’s exact test *p*-value= 0.02003; OR = 1.38; 95% CI (1.04, 1.81)).

Examining the larger study of Ouwens et al. [4], the list of FDR-significant heritable genes were not enriched for either the longevity or ageing-associated gene lists.

### 3.6. Candidate Non-Heritable Genes

From the results of gene heritability analysis, a list of 460 genes that potentially are not significantly heritable was compiled, based on the lowest FDR values (Appendix A).

The Gene2Func procedure in FUMA was also conducted on this set of 460 non-heritable genes, which showed no significantly enriched pathways or enrichment for GWAS results. Tissue-specific analysis for the 30 general tissues also indicated this gene set is not significantly enriched in any of the tissues studied (Appendix A).

The OATS non-heritable list of genes (*n* = 460) was also compared with the prior published study based on microarray data of Wright et al. [7] to assess if any of these genes were significantly heritable in that study. Of the 220 of the 460 genes available, except for one gene *KLRK1*, none of the other genes were significant, lending support for the OATS non-heritable list of genes having no significant heritability.

### 3.7. Cross-Study Comparison of Gene Expression Heritability across Three Studies

We compared gene expression heritability results across three studies, using OATS (Illumina array) and two other partially independent published studies (60% participant overlap): one was array-based (Affymetrix array) [7] and the other a RNA sequencing study [4]. Since array probes and RNA seq transcripts cannot be matched, we based the analysis on Ensembl gene IDs [21]. Initially, for the OATS data, when there was more than one probe for a single gene, and the probe with the highest heritability was selected, similarly to the method used by Wright et al. [7]. The OATS data had 19,256 genes, while that of Wright et al. [7] included 18,392 genes. In contrast, for the third set of data, Ouwens et al. [4], transcripts were chosen based on the selection criteria as described by the authors, resulting in a total of 11,409 genes.

Overlapping genes across the three studies were then used for the analysis (N = 9479 genes). This set of genes had an average heritability value of 0.241 (SD ± 0.162) [OATS], while the average heritability values of Ouwens et al. and Wright et al. were 0.264 (SD ± 0.155) and 0.133 (SD ± 0.145), respectively (Appendix A). In addition, the heritability values of the OATS data for this set of overlapping genes were highly correlated with that of other two studies (r~0.22 to 0.32) (Appendix A). Appendix A, respectively, show the Manhattan plot and the scatter plots comparing heritabilities for these cohorts using the common 9479 genes. In addition, the observed vs. expected *p*-value distributions (QQ plot) for the three studies are shown in Appendix A. There were 76 FDR-significant heritable genes for OATS, 1027 for Wright et al. [7], and 2012 for Ouwens et al. [4].

However, only 38 FDR-significant genes were common across all three studies (Figure 2, Appendix A). Table 4 shows the top 10 of the 38 significantly heritable genes for OATS and the corresponding h^2^ values in the other two studies. Genes such as *ERAP2* (h2 range 0.85–0.88), *LILRA3* (h^2^ range 0.75–0.85), *MYOM2* (h^2^ range 0.72–0.87), and *CLEC12A* (h^2^ range 0.74–0.90) were highly heritable in all three studies.

Gene set analyses wase performed on FDR-significant overlapping heritable genes found across the three datasets (*n* = 38) using the Gene2Func procedure in FUMA. There are two significant canonical pathways. However, there are no significant GO biological processes. The significant, canonical pathways, Hallmark gene sets, immunological signature, and cancer modules are shown in Appendix A. The genes in the two canonical pathways are related to the immune response. Tissue specific analysis indicated that this gene set is significantly enriched for expression in the blood (Appendix A). FUMA GWAS catalogue results indicated no significant gene sets.

We then checked if any of these 38 common heritable genes overlap with the set of ageing-associated (*n* = 265) and longevity genes (*n* = 225). There were no overlapping genes with the set of ageing-associated genes, and only one gene *HP* (Haptoglobin located on chr 16) overlapped with longevity genes. This gene showed high heritability in OATS sample (h^2^ = 0.63; FDR= 3.01 × 10^−2^).

## 4. Discussion

The current study is the first study reporting the heritability of gene expression from peripheral blood in both older men and women. We observed significant heritability for 5269 probes (19.85%) from 4603 unique genes. Many of the heritable genes were involved in immune-related biological processes. A cross study comparison revealed a small set of 38 genes that were significantly heritable and were in general involved in the immune response.

In our analysis, approximately 20% of the available 26537 probes were significantly heritable. The highest heritability (h^2^ = 0.87) was observed for an *ERAP2* transcript, while the lowest heritability (h^2^ = 0.27) was observed for a *BRD9* transcript. Coincidentally, both the highest and the least heritable significant genes were located on chromosome 5. The *ERAP2* gene encodes for Endoplasmic Reticulum Aminopeptidase 2, which plays a main role in the shortening of peptides during the generation of HLA class I-binding peptides, hydrolysing the basic residues of arginine and lysine [24,25]. This gene has been implicated in many GWAS (*n* = 36) for a wide range of phenotypes ranging from blood protein levels and blood pressure to bowel and chronic inflammatory diseases. The second highest heritability was observed for the transcript of the gene *NQO2*, which encodes for a quinone reductase and is a member of the thioredoxin family. Mutations of this gene have been associated with many neurodegenerative diseases and several cancers including breast cancer [26,27]. The gene with lowest significant heritability, *BRD9* encodes for a bromodomain-containing protein 9, which plays a main role in chromatin remodelling and regulation of transcription.

Nearly 24% of the total 19,256 genes were significantly heritable (Appendix A). Of these significant genes, 576 had multiple heritable probes. The average gene heritability (h^2^ = 0.40) of the current study is higher compared to that of other prior published studies, which ranged from 0.10 to 0.32 [4,5,6,7,8]. This may be due to differences in the methodologies used for the heritability analysis and the age of the participants (e.g., 18–65 years aged twins were studied by Wright et al. [7] and families aged 18–85 years by Price et al. [6]).

A gene set analysis of the top 10% significantly heritable genes indicated enrichment for genes related to the immune response, which is not unexpected given that gene expression in whole blood was measured. Similar results were reported in prior studies [4,7]. Tissue-specific analysis in 30 general tissues indicated that this gene set is significantly enriched not only in blood but also in two other tissues (spleen, lung).

Although gene length showed no significant correlation with the heritability of gene expression of the OATS sample, the percentage GC content showed a significant positive relationship, suggesting that as the percentage GC content of a gene increases, its heritability also increases. These GC content results are in accordance with the previously published results of Ouwens et al. and Loh et al. [4,28]. We did not consider the study of Wright et al. [7], given methodological differences in their heritability analyses. However, as the percentage GC content affects complex pathways including DNA methylation, recombination [29] and replication timing [30], further studies are required to understand its role in heritability of gene expression.

Our set of heritable genes was significantly enriched for both longevity- and ageing-associated genes, indicating the importance of further investigating the genetic factors affecting the expression of these genes. Genes that have been previously reported to be associated with ageing and neurodegenerative disease, such as *BDNF* (h^2^ = 0.27; FDR = 0.037) and *FOXO3* (h^2^ = 0.31; FDR = 0.024), showed low but still significant heritability in the OATS sample, whereas Wright et al. [7] reported no significant heritability for the *BDNF* gene, although *FOXO3* showed significant heritability. Interestingly, *APOE,* an Alzheimer’s disease-related gene, was not significantly heritable in our sample, nor in the results of Wright et al. [7]. When we examined enrichment for longevity- and ageing-associated genes in the heritable results of a larger sample of individuals from across the adult lifespan but with a younger mean age (36.7 years) (Ouwens et al., 2020 [4]), there was no enrichment for either ageing or longevity-related genes. These results suggest that heritable genes in older adults are enriched for ageing- and longevity-related genes.

The lack of significant heritability for a number of genes suggests environmental factors, such as diet and lifestyle, play a major role in determining levels of gene expression. Our final list of the genes that do not have significant heritability provides impetus for further investigation of environmental factors that influence their expression in older age. The list includes *ELOVL2*, which is involved in fatty acid metabolism and is differentially methylated across the lifespan and is included in many epigenetic clocks, e.g., [31,32]. Three olfactory receptor genes are included (*OR1B1, OR3A2, OR2T6*), that are involved in smell perception. In addition, the list not only includes protein-coding genes but also non-coding RNAs, such as microRNAs, scaRNAs, and long non-coding RNAs. The complex interactions between expression of these genetic loci with various environmental factors is an important area for future study.

A combined analysis across three studies, OATS, and data from two partially independent studies [4,7] revealed a total of 9479 genes that were common across the three studies. However, only 38 genes (0.4%) were significantly heritable in all cohorts. Genes such as *ERAP2* (h^2^~0.85–0.88), *LILRA3* (h^2^~0.75–0.85), *MYOM2* (h^2^~0.72–0.87), and *CLEC12A* (h^2^~0.74–0.90) were highly heritable in all three studies, indicating the importance of further study of these genes. *LILRA3* is located on chromosome 19 and encodes a member of a family of immunoreceptors that are expressed predominantly in monocytes and B cells. The gene *MYOM2,* located on chromosome 8, encodes a protein that plays a main role in myofibrillar M bands. The gene *CLEC12A*, on chromosome 12, encodes a member of the C-type lectin superfamily, which plays a main role in cell adhesion and cell–cell signalling.

A functional analysis of the 38 overlapping genes between studies indicated enrichment for genes from immunological trait pathway. Although this set of genes showed no overlap with ageing-associated genes, one longevity-related gene, *HP* (Haptoglobin located on chr 16), was observed. Previous studies have reported an association of *HP* gene variants with human longevity [33,34], indicating the importance of further investigating this gene. The *HP* gene product binds to plasma haemoglobin and has anti-inflammatory and immune functions [33] and its expression showed high heritability in OATS. Another interesting gene observed in this set of 38 genes is *PEX6*, which is located on chromosome 6, and is highly heritable in our OATS analysis (h^2^ = 0.83). The *PEX6* missense variant, rs1129187, has been associated with Alzheimer’s disease in *APOE* ε4 carriers [35]. Other studies have reported *PEX6* mutations to be associated with Peroxisome biogenesis disorders, along with hearing impairment and retinitis pigmentosa [36,37].

At an individual study level, Ouwens et al. reported the most FDR-significant genes, followed by Wright et al. and then this study. The small number of overlapping heritable genes across the cohorts may be due to differences in the methodology (array, sequencing, data QC, analysis, sample age, sample size, number of genes assayed). The mean age of the Dutch studies was less than 40 years old (Wright et al. [7], 32 years; Ouwens et al., [4] 36.7 years) while the current study is much older with a mean of 76 years. Given the minimal overlap in age across studies it highlights the importance of examining the heritability of gene expression at different stages of the lifespan. This is backed up by previous research, where Vinuela et al. [38] demonstrated age-dependent changes in gene expression in a large sample of female twins, which was driven by a complex mixture of environmental and genetic factors [38]. This indicates the importance of studying age-related changes in gene expression and its regulation in larger samples of older adults. Although some of the previous reports studied heritability in both sexes, this is the first study to report heritability in both men and women specifically in older adults. Although our heritability results did not show any sex-related differences, given our modest sample size, it will still be important to investigate sex differences in future studies.

Limitations of the current study should be considered. Due to the relatively small sample size, the current study was most likely underpowered. However, the sample size used in this study is much larger than the previous study of older adults [9]. As gene expression was assessed using an array, the number of transcripts studied was limited compared to RNA sequencing. Although we compared our results with that of Wright et al. and Ouwens et al., there were a number of limitations including methodological differences such as the assay (e.g., Illumina vs. Affymetrix arrays vs. RNA sequencing) and the age range. Furthermore, this analysis was also not ideal given the overlap in samples and hence only partial independence between the Wright et al. and Ouwens et al. studies. Gene expression in individual WBC subtypes was not assessed and may have confounded the results. Although total cell count was added as a covariate, the frequencies of the lymphocyte subtypes such as cytotoxic CD4T cells were not controlled in this analysis. An additional limitation is that all the individuals examined in this study were of European descent, and our results may not apply to other ethnic/racial groups.

In conclusion, our study in older adults suggests a large number of heritable genes, with little overlap with studies that examined younger ages. Although our study requires replication in other independent cohorts of older adults, it suggests that there are age-related differences in gene expression and that it is important to specifically examine these questions in older age groups. Our work also suggests a small list of genes that potentially lack significant heritability that may be good candidates for examination of the influence of environmental factors and/or epigenetic factors on their expression.

## Figures and Tables

**Figure 1 genes-15-00495-f001:**
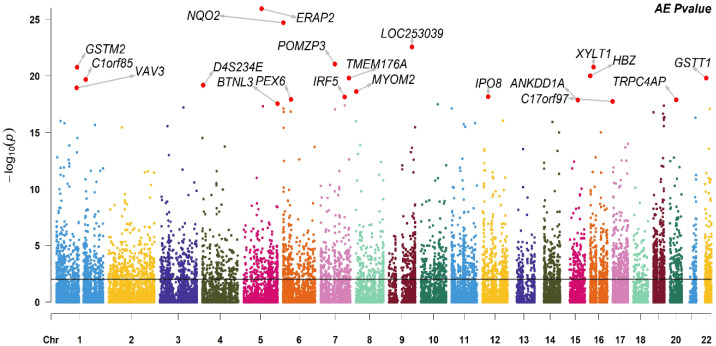
Manhattan plot of the *p*-values for gene heritability (N = 19,256), with the top 20 FDR-significant heritable genes indicated. The horizontal line indicates threshold of FDR significance. The probes on each chromosome are indicated in different colours.

**Figure 2 genes-15-00495-f002:**
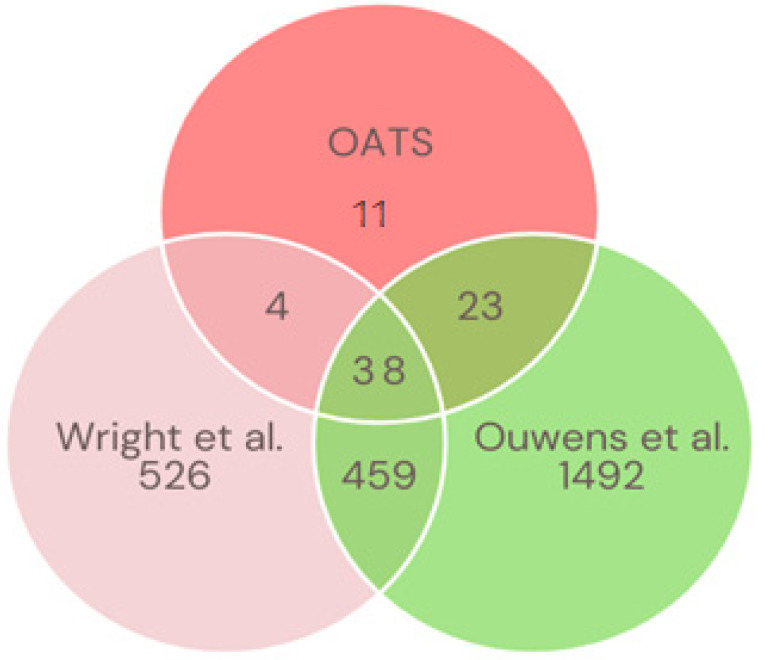
Comparison of significantly heritable genes across three studies (current OATS study, Wright et al., 2014 [7], and Ouwens et al., 2020 [4]) using only overlapping genes common to all cohorts. The Venn diagram shows that there were 38 genes that were significantly heritable across all studies.

**Table 1 genes-15-00495-t001:** Demographic details of the OATS twin sample used for heritability analysis.

Variable	Total Sample	MZ	DZ	*p*-Value
N	246	142 (71 pairs)	104 (52 pairs)	N/A
Sex—Female, N (%)	172 (69.91)	102 (71.83)	70 (67.31)	0.402
Age (yrs) Mean (SD)	75.79 (5.44)	75.60 (5.49)	76.04 (5.39)	0.537
Age Range (yrs)	(69.4–93.5)	(69.7–93.5)	(69.4–93.1)	N/A
BMI (Kg/m^2^)	27.63 (4.74)	27.92 (4.89)	27.22 (4.52)	0.764
WBC count (K/μL)	6.41 (1.66)	6.33 (1.59)	6.51 (1.73)	0.392
Current smoker	8 (3.25%)	5	3	NA
Alcohol user (>30 drinks per month)	77 (31.30%)	46	31	NA

Notes: MZ—monozygotic twins; DZ—dizygotic twins; SD—standard deviation. Differences between MZ and DZ groups were evaluated using *t*-tests (continuous variables) and chi-squared tests (categorical variables). N/A—not applicable (statistical test not performed). The tests of significance were not conducted for smoking and alcohol users’ status. *p*-values were obtained based on 5000 permutations.

**Table 2 genes-15-00495-t002:** Top 10 FDR-significant heritable probes in OATS.

Probe ID	Gene Symbol	Chr	h^2^	CI	Bootstrap CI	FDR
ILMN_1743145	*ERAP2*	5	0.87	0.81–0.91	0.84–0.90	2.91 × 10^−22^
ILMN_1712918	*NQO2*	6	0.86	0.80–0.91	0.82–0.91	2.54 × 10^−21^
ILMN_3236498	*LOC253039*	9	0.85	0.78–0.90	0.81–0.89	2.35 × 10^−19^
ILMN_1805377	*POMZP3*	7	0.81	0.73–0.87	0.76–0.87	5.81 × 10^−18^
ILMN_1830462	*XYLT1*	16	0.85	0.77–0.90	0.80–0.89	7.66 × 10^−18^
ILMN_2201580	*GSTM2*	1	0.8	0.71–0.86	0.74–0.87	7.66 × 10^−18^
ILMN_1713458	*HBZ*	16	0.78	0.69–0.85	0.73–0.84	3.74 × 10^−17^
ILMN_1791511	*TMEM176A*	7	0.81	0.72–0.87	0.76–0.86	4.58 × 10^−17^
ILMN_1730054	*GSTT1*	22	0.8	0.71–0.86	0.75–0.86	4.58 × 10^−17^
ILMN_1698243	*C1orf85*	1	0.8	0.71–0.86	0.75–0.84	5.45 × 10^−17^

Notes: Probe ID—Illumina array probe ID; h^2^—heritability; Chr—chromosome; FDR—false discovery rate; *ERAP2*—Endoplasmic Reticulum Aminopeptidase; *NQO2*—N-Ribosyldihydronicotinamide—quinone reductase 2; *LOC253039; PSMD5-AS1* PSMD5 antisense RNA 1 (head-to-head); *POMZP3—POM121* and *ZP3* Fusion; *XYLT1*-Xylosyltransferase 1; *GSTM2*—Glutathione S-Transferase Mu 2; *HBZ*—Haemoglobin Subunit Zeta; *TMEM176A*—Transmembrane Protein 176A; *GSTT1*—Glutathione S-Transferase Theta; *C1orf85*—Chromosome 1 Open Reading Frame 85 (Glycosylated Lysosomal Membrane Protein).

**Table 3 genes-15-00495-t003:** The 10 most significant GO biological processes and canonical pathways for the top FDR-significant 10% heritable genes (*n* = 460).

GO Biological Processes/Canonical Pathways	N Genes in Pathway	N Genes Overlap	FDR
REACTOME_NEUTROPHIL_DEGRANULATION	389	51	7.82 × 10^−16^
REACTOME_INNATE_IMMUNE_SYSTEM	826	66	2.17 × 10^−10^
REACTOME_ANTIMICROBIAL_PEPTIDES	56	13	7.59 × 10^−6^
GOBP_ANTIMICROBIAL_HUMORAL_RESPONSE	80	16	1.10 × 10^−5^
GOBP_IMMUNE_RESPONSE	1237	73	3.81 × 10^−5^
GOBP_ANTIMICROBIAL_HUMORAL_IMMUNE_RESPONSE_MEDIATED_BY_ANTIMICROBIAL_PEPTIDE	53	12	9.82 × 10^−5^
GOBP_DEFENSE_RESPONSE	1297	73	1.27 × 10^−4^
GOBP_DEFENSE_RESPONSE_TO_BACTERIUM	212	23	1.27 × 10^−4^
GOBP_INFLAMMATORY_RESPONSE	645	45	1.27 × 10^−4^
GOBP_HUMORAL_IMMUNE_RESPONSE	168	19	7.2 × 10^−4^

**Table 4 genes-15-00495-t004:** Top 10 of the 38 significant genes overlapped across three heritability studies (OATS, Wright et al., 2014 [7], and Ouwens et al., 2020 [4]).

Gene Symbol	Probe ID	Chr	OATS		Wright		Ouwens	
			h^2^	FDR	h^2^	FDR	h^2^	FDR
*ERAP2*	ENSG00000164308	5	0.87	8.94 × 10^−4^	0.88	3.53 × 10^−25^	0.85	1.95 × 10^−28^
*NQO2*	ENSG00000124588	6	0.86	4.64 × 10^−5^	0.59	3.21 × 10^−8^	0.87	2.25 × 10^−37^
*XYLT1*	ENSG00000103489	16	0.85	3.14 × 10^−5^	0.41	6.56 × 10^−3^	0.43	2.58 × 10^−3^
*PEX6*	ENSG00000124587	6	0.83	1.32 × 10^−5^	0.32	3.21 × 10^−2^	0.88	1.19 × 10^−40^
*TMEM176A*	ENSG00000002933	7	0.81	7.35 × 10^−3^	0.61	2.50 × 10^−13^	0.91	3.76 × 10^−45^
*LILRA3*	ENSG00000170866	19	0.81	4.64 × 10^−5^	0.75	8.76 × 10^−15^	0.85	1.08 × 10^−25^
*NSG1 (D4S234E)*	ENSG00000168824	4	0.81	1.50 × 10^−4^	0.35	1.33 × 10^−2^	0.82	5.02 × 10^−24^
*ANKDD1A*	ENSG00000166839	15	0.80	8.94 × 10^−4^	0.34	2.59 × 10^−2^	0.43	7.75 × 10^−4^
*CFD*	ENSG00000197766	19	0.80	1.85 × 10^−4^	0.57	1.23 × 10^−9^	0.74	6.82 × 10^−11^
*MYOM2*	ENSG00000036448	8	0.79	4.80 × 10^−3^	0.72	3.01 × 10^−20^	0.87	1.20 × 10^−32^

Notes: h^2^—heritability; Chr—chromosome; FDR—false discovery rate; Wright—Wright et al. [7]; Ouwens—Owens et al. [4]. *ERAP2*—Endoplasmic Reticulum Aminopeptidase 2; *NQO2*—N-Ribosyldihydronicotinamide: quinone reductase 2; *XYLT1*—Xylosyltransferase 1; *PEX6*—Peroxisomal Biogenesis Factor 6; *TMEM176A*—Transmembrane Protein 176A; *LILRA3*—Leukocyte Immunoglobulin-Like Receptor A3; *NSG1*—Neuronal Vesicle Trafficking-Associated 1; *ANKDD1A*—Ankyrin Repeat And Death Domain-Containing 1A; *CFD*—Complement Factor D; *MYOM2*—Myomesin 2.

## Data Availability

The authors confirm that the data supporting the findings of this study are available within the article and its Appendix A. Additional data that support the findings of this study are available from the corresponding author upon request.

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
