# Peer review of "Heritability of Gene Expression Measured from Peripheral Blood in Older Adults"

_genes, 2024, doi:10.3390/genes15040495_

Round 1

Reviewer 1 Report

Comments and Suggestions for Authors

Sri C. Kanchibhotla et al described their work clearly and exhaustively. Given the immune context, I suggest, for future works, studying the different gene expression in several pro-inflammatory contexts and analyzing, if present, the different response in gene expression. Other this suggestion the work is well done. The only negative note is the writing font on line 428-432, it must be consistent with the rest of the text.

Author Response

Comments from the reviewer(s):

Reviewer #1 (Comments to the Author):

Sri C. Kanchibhotla et al described their work clearly and exhaustively. Given the immune context, I suggest, for future works, studying the different gene expression in several pro-inflammatory contexts and analyzing, if present, the different response in gene expression. Other this suggestion the work is well done. The only negative note is the writing font on line 428-432, it must be consistent with the rest of the text.

Response: The authors appreciate the feedback provided by the reviewer and spotting this error in the font.  As the reviewer suggested, the font on lines 428-432 is now changed in according to the manuscript font.

Reviewer 2 Report

Comments and Suggestions for Authors

Summary:

In this study by Kanchibhotla et al., titled "Heritability of Gene Expression Measured from Peripheral Blood in Older Adults," the authors analyzed the heritability of blood gene expression from The Older Australian Twin Study (OATS) cohort. They identified 4603 significantly heritable genes, enriched in immune response and aging regulation pathways. Among these, ERAP2 was found to be the most heritable, while BRD9 was the least heritable. They also found that the heritability of genes is significantly positively correlated with GC percentage. The authors noted limited correlation between their study and two previous studies (Wright, 2024; Ouwens, 2020) and attributed this to differences in the age composition of the study cohort and methodology.

General comment:

In general, I believe the evidence provided by the authors is sufficient to support the conclusion of this manuscript, also providing resources for other human genetic studies. The discrepancy between this study and previous studies is concerning but may be attributed to a different cohort. Additionally, there are some formatting errors that need to be fixed.

Specific questions and comments:

1.     In section 3.3, the author mentioned no significant correlation between gene length and expression heritability. A previous report (Vinuela, 2018) suggests aging has an effect on splicing. Although such analysis is understandably impossible with microarray data, the author could analyze the correlation between average intron number and heritability.

2.     “3.5. Ageing and Longevity Enrichment Analysis” has formatting issue, I believe the first paragraph of 3.6 belongs here.

3.     What is the author's purpose and conclusion for the aging/longevity enrichment analysis? Are they suggesting that aging regulators have more heritability in this aged cohort? The author should compare their results with previous studies performed on the young/general population to see if the enrichment of aging-associated/longevity-related genes is different. This will also help demonstrate the discrepancy between their results and previous reports could be attributed to age.

4.     In section 3.6, “a list of 100 genes that potentially are 268 not significantly heritable was compiled, based on the lowest FDR values”, are the authors choosing all genes with an FDR of 1? 100 genes is quite a low number for pathway analysis. The author could try to expand the gene list and see if this yields any significant pathways.

5.     Similarly: “Gene-set analyses was performed on FDR significant overlapping heritable genes 319 found across the three datasets (n=38)”, with only 38 genes, it is very challenging to perform any meaningful pathway analysis (line 319).

6.     line 466, I believe it should be Wright et al., 2014.

Author Response

Reviewer #2 (Comments to the Author):

General comment:

In general, I believe the evidence provided by the authors is sufficient to support the conclusion of this manuscript, also providing resources for other human genetic studies. The discrepancy between this study and previous studies is concerning but may be attributed to a different cohort. Additionally, there are some formatting errors that need to be fixed.

Specific questions and comments:

  1. In section 3.3, the author mentioned no significant correlation between gene length and expression heritability. A previous report (Vinuela, 2018) suggests aging has an effect on splicing. Although such analysis is understandably impossible with microarray data, the author could analyze the correlation between average intron number and heritability.

Response: As suggested by the reviewer we have examined the correlations of heritability with number of introns and exons in a gene using the data provided in Piovesan et al., 2019 [1]. The correlations of heritability with average intron and exon numbers were -0.0482 (p-value=0.00245) and -0.0499 (p-value=0.0017), respectively. Although these correlations are significant due to the large number of genes involved, the size of these correlations are very small. Moreover, the number of exons and introns are highly correlated with gene length (r= 0.33 & 0.32 respectively), but gene length was not significantly correlated with heritability (see lines 243 – 244). Given the very small correlations observed and the inconsistency of these relationships we have not included the correlations of heritability with gene introns / exons in the manuscript.

  1. “3.5. Ageing and Longevity Enrichment Analysis” has formatting issue, I believe the first paragraph of 3.6 belongs here.

Response: We thank the reviewers for pointing out the error. The first paragraph of section 3.6 is now moved to section 3.5.

  1. What is the author's purpose and conclusion for the aging/longevity enrichment analysis? Are they suggesting that aging regulators have more heritability in this aged cohort? The author should compare their results with previous studies performed on the young/general population to see if the enrichment of aging-associated/longevity-related genes is different. This will also help demonstrate the discrepancy between their results and previous reports could be attributed to age.

Response: We have now performed the analysis as suggested by the reviewer. We examined a large heritability study by Ouwens et al, that used a sample from across the lifespan with a younger mean age compared to our study and there was no enrichment for the longevity or the ageing-associated gene lists. These results suggest that heritable genes in older adults are enriched for ageing and longevity-related genes. We have updated the manuscript (lines 265-270 and 397-401) and the Supplementary Tables (Supplementary Table 4d and Supplementary Table 4e).

  1. In section 3.6, “a list of 100 genes that potentially are 268 not significantly heritable was compiled, based on the lowest FDR values”, are the authors choosing all genes with an FDR of 1? 100 genes is quite a low number for pathway analysis. The author could try to expand the gene list and see if this yields any significant pathways.

Response: There were in fact 7180 genes with FDR=1 in our data. The bottom 100 non-heritable genes were compiled based on heritability values. As suggested by the reviewer, we have now extended the list to bottom 460 genes to match the number of top 10% FDR significant genes used for pathway analysis (Supplementary Table 5). We have now performed the gene set analysis using FUMA and found no tissue specific enrichment or significant pathways for this set (Supplementary Figure 3). Also, we have now included the gene-heritability results for the full set of genes examined in this manuscript (Supplementary Table 9) for the readers to extract the heritable and non-heritable genes based on the thresholds of their choice.

  1. Similarly: “Gene-set analyses was performed on FDR significant overlapping heritable genes 319 found across the three datasets (n=38)”, with only 38 genes, it is very challenging to perform any meaningful pathway analysis (line 319).

Response: The authors agree with the reviewer’s opinion. Although 38 is a small number for any pathway analysis, these 38 genes were significantly overlapping across the three datasets and hence we wanted to examine if this set of genes intersects with any well-known pathway gene sets.

  1. line 466, I believe it should be Wright et al., 2014.

Response: We thank the reviewer for spotting this. It is a typographical error. In the latest version of the manuscript, it is changed to 2014.

References:

  1. Piovesan, A.; Antonaros, F.; Vitale, L.; Strippoli, P.; Pelleri, M.C.; Caracausi, M. Human protein-coding genes and gene feature statistics in 2019. BMC Res Notes 2019, 12, 315, doi:10.1186/s13104-019-4343-8.
  2. Ouwens, K.G.; Jansen, R.; Nivard, M.G.; van Dongen, J.; Frieser, M.J.; Hottenga, J.J.; Arindrarto, W.; Claringbould, A.; van Iterson, M.; Mei, H.; et al. A characterization of cis- and trans-heritability of RNA-Seq-based gene expression. European journal of human genetics : EJHG 2020, 28, 253-263, doi:10.1038/s41431-019-0511-5.
